# Effect of Different Adjuvants on the Immunogenicity of a Recombinant Herpes Zoster Vaccine in Mice, Rats and Non-Human Primates

**DOI:** 10.3390/vaccines13111124

**Published:** 2025-10-31

**Authors:** Xiaoyang Liu, Shaohua Gong, Jingyi Xu, Ying Wei, Xuyan Chen, Yucheng Wu, Zhengli Zhang, Junyu Ma, Yili Yang, Shuhua Tan

**Affiliations:** 1Shanghai Wisdom Biotechnology Co., Ltd., Shanghai 200120, China; 2Jingsu Wisdom Biotechnology Co., Ltd., Taizhou 225300, China; gongshaohua@liweiside.com (S.G.); xujingyi@liweiside.com (J.X.); 18861016982@163.com (Y.W.); xiaochengzi@liweiside.com (X.C.); wuyucheng@schbio.com (Y.W.); zhangzhengli@liweiside.com (Z.Z.); 3Department of Cell and Molecular Biology, School of Life Science and Technology, China Pharmaceutical University, Nanjing 211100, China; majunyu@liweiside.com; 4International Centre for Genetic Engineering and Biotechnology, Taizhou 225300, China; yangliyi@icegb.cn

**Keywords:** varicella zoster virus (VZV), compound adjuvant, MF59, CpG1018, cell-mediated immunity (CMI)

## Abstract

**Background**: Varicella zoster virus (VZV) is a globally circulating pathogen that usually infects children and establishes a latent state in host nerve cells. Recurrence of latent varicella zoster virus (VZV) is often triggered by predisposing factors such as aging and immune dysfunction, which may lead to herpes zoster (HZ) and its related complications. At present, there is no specific treatment for herpes zoster or postherpetic neuralgia, so vaccination is an important preventive measure. **Methods**: In this study, a variety of vaccine formulations were developed by combining the gE protein with different adjuvants. Enzyme-linked immunosorbent assay (ELISA), flow cytometry, and ELISpot were used to evaluate the immune response induced by each combination of vaccines in C57BL/6 mice, and the optimal combination of adjuvants. Then, its immunogenicity was verified in SD rats and rhesus monkeys. **Results**: All combinations of gE/squalene oil-in-water emulsion (SWE)/CpG1018 adjuvant induced a good humoral immune response 28 days after secondary immunization. GE/SWE/CPG1018, combined with adjuvant, induced a higher cellular immune response in mice. The selected gE/SWE/CpG1018 combined with the adjuvant vaccine combination could effectively stimulate the humoral and cellular immune responses in SD rats and rhesus monkeys. **Conclusions**: The gE/SWE/CpG1018 combined with adjuvant vaccine may be a low-cost and highly effective vaccine candidate for the prevention of varicella zoster.

## 1. Introduction

Varicella zoster virus (VZV), a highly contagious double-stranded DNA neurotropic human α-herpesvirus, encompasses 71 genes that code for 67 distinct proteins, among which are six glycoproteins, namely gE, gB, gH, gI, gC, and gL [1]. As the initial episode of varicella infection begins to fade, VZV enters a latent phase, lying dormant in the posterior root ganglia of the spinal cord and the sensory ganglia of the cranial nerves. In the elderly or those with weakened immune systems due to old age or immunosuppression, VZV cannot be effectively contained, ultimately leading to the onset of herpes zoster (HZ) [2].

Vaccination against HZ is currently the most effective preventive approach. The two principal HZ vaccines available on the market nowadays are Zostavax^®^ (Merck, NJ, USA) and Shingrix^®^ (GSK, London, UK) [3]. Zostavax^®^ is a live attenuated vaccine, which has demonstrated good efficacy in reducing the incidence of both Postherpetic neuralgia (PHN) and HZ [4,5,6]. Shingrix^®^, an adjuvanted subunit vaccine manufactured by GlaxoSmithKline (GSK), has an advantage over the attenuated vaccine Zostavax^®^ in terms of both safety and efficacy. The recombinant subunit vaccine is composed of the VZV-gE protein and the AS01 adjuvant. VZV gE serves as the major antigen for inducing neutralizing antibodies, containing B-cell and CD4^+^ T-cell epitopes [7]. The AS01 adjuvant consists of QS21 and monophosphoryl lipid A (MPL) [8]. The synergistic interaction between QS21 and MPL bolsters the adaptive immune response of the AS01 adjuvant, inducing a Th1-type cellular immune response and facilitating the production of higher levels of cytokines [9]. However, as the core component of AS01 adjuvant, the limited supply of QS21 has become one of the key factors hindering the wide popularization of vaccines [10]. The high price of the Shingrix vaccine has become a disadvantage that cannot be ignored, especially in the context of the rapid growth of demand in the domestic market. In view of this situation, the development of an economical and highly safe herpes zoster vaccine is particularly urgent.

Adjuvants are substances added to vaccines to enhance high-purity antigens with insufficient immune stimulation ability [11]. MF59 is an oil-in-water emulsion adjuvant with triterpene squalene as the oil phase and consists of squalene, stable nonionic surfactants Tween 80 and Span85 [12]. MF59 can enhance the recruitment and activation of antigen-presenting cells (APCs), accelerate antigen uptake, promote cell migration to lymph nodes, enhance immunogenicity and cross-protection [13], and was originally used to improve the immune efficacy of influenza vaccine in the elderly population [14]. The CD4-independent adjuvant effect of MF59 may help to enhance the effectiveness of vaccines in the elderly and immunocompromised patients [15], and MF59 adjuvant-containing influenza vaccines have been approved for market in more than 30 countries around the world. During the COVID-19 pandemic, results from a randomized, double-blind, placebo-controlled phase I clinical trial of a COVID-19 subunit vaccine in Australia showed that after administration of a vaccine formulation containing the MF59 adjuvant, spike-specific CD4^+^ T cell responses and multifunctional helper T cell (Th1 and Th2) responses were enhanced. The MF59 adjuvant used in this study is the squalene oil-in-water emulsion adjuvant produced by Seppic, which is described by SWE in the experimental results section.

CpG is a kind of unmethylated cytosine-guanine dinucleotide sequence, which is commonly found in the bacterial genome and has the ability to activate the innate immune pattern recognition receptor-Toll-like receptor (TLR) 9 [16]. Artificially modified and synthetic CpG oligodeoxynucleotides (ODNs) have a variety of functions, such as triggering innate immune responses and protecting the host from pathogens. They can enhance the secretion of cytokines and enhance the specific immune response after vaccination [17]. They also promote the activation of natural killer (NK) cells and cytotoxic T lymphocytes (CTL) for the treatment of cancer [18]. CpG1018, a B-grade CpG ODN widely used in vaccine research, activates TLR9 receptors. Human B cells and plasmacytoid dendritic cells highly express TLR9 in response to CpG stimulation, thereby initiating an immunostimulated cascade. This eventually leads to the maturation, differentiation, or proliferation of NK cells, T cells, monocytes, and macrophages [19,20]. The Heplisav-B vaccine containing CpG1018 has been approved by the FDA. Clinical studies have shown that the vaccine provides protection of up to 90% [21].

Combinations of different adjuvant types can effectively integrate all components and trigger both Th1 and Th2 immune responses. For example, when VZV gE protein was paired with a composite adjuvant such as MF59 + QS-21 or MF59 + CpG, mice produced antibody levels and CD4^+^ T cell immune responses comparable to those induced by Shingrix^®^ [22]. In this study, we combined different adjuvant combinations with gE proteins to formulate different vaccine combinations for mouse administration. The level of immune response was assessed 28 days after the secondary immunization to screen for an adjuvant combination that could enhance specific T cell immune responses. The vaccine composition identified in this study can specifically enhance the cellular immune response against varicella zoster virus. It can significantly increase the expression of cytokines such as INF-γ and IL-2 in CD4^+^ T cells of mice, improve humoral and cellular immune responses, and show high immune activity to prevent varicella zoster virus infection in clinical practice. The vaccine composition selected in this study is inexpensive and easily available, which can effectively promote vaccine production and reduce production costs.

## 2. Materials and Methods

### 2.1. Vaccine Preparation

The preferred codon optimization algorithm for CHO cells was used to optimize and synthesize the gE target gene. The pLK002-GS vector was constructed to express the recombinant protein, and it was transfected into CHO-K1BN cells to obtain stable and highly expressed cell lines. After obtaining the monoclonal cell lines, cell culture and protein expression, purification was carried out to finally obtain the gE protein. Different adjuvants were combined with the gE protein to form different vaccine combinations. The gE protein was diluted to 5 μg per mouse for injection, mixed with the adjuvants, and then diluted to 50 μL with phosphate-buffered saline (PBS, pH 7.5) for injection into mice. In this study, three adjuvant formulations were selected as the main research objects, including SWE (Seppic, Paris, France), CpG1018 (AsymBio, Beijing, China) and aluminum hydroxide (Croda, Frederikssund, Denmark), and the adjuvant AS01_B_ used in the marketed herpes zoster vaccine, Shingrix^®^ (GSK, London, UK), was used as the positive control group to evaluate the effects of different adjuvant formulations on the humoral immune response in mice. To screen out the adjuvant combinations that can enhance the specific T-cell immune response. We selected 120 C57BL/6J mice and randomly divided them into 20 groups, with 6 mice in each group. Among them, 9 groups received injections of single adjuvant groups (Table 1), and 11 groups received injections of combined adjuvant groups (Table 2). Muscle injections were performed every four weeks, and a total of two immunizations were conducted. The specific vaccination schedule is shown in Table 1. In the single adjuvant group, the S1 group was a PBS negative control group, the S2–S9 groups were treated with 5 μg gE protein, the S2 group was treated with AS01_B_ adjuvant as a positive control group, the S3–S5 groups were treated with different doses of SWE adjuvant, and the S6–S9 groups were treated with different doses of CpG1018 adjuvant. The effects of different doses of SWE and CpG1018 on immunogenicity were investigated. In the compound adjuvant groups, the C1 group was a PBS negative control group, the C2–C11 groups were treated with 5 μg gE protein, the C2 group was treated with AS01_B_ adjuvant as a positive control group, and the C3–C5 groups were treated with 50 μg aluminum hydroxide adjuvant and different doses of CpG1018 adjuvant. The immunogenicity of aluminum hydroxide combined with different doses of CpG1018 was observed. In the C6–C8 and the C9–C11 groups, 25 μL/12.5 μL SWE adjuvant was added, followed by different doses of CpG1018 adjuvant to observe the dose–response relationship between adjuvant SWE and CpG1018.

### 2.2. Mouse Study

#### 2.2.1. Mice Feeding and Serum Treatment

Specific Pathogen-Free (SPF) female C57BL/6J mice (8 weeks old) were provided by Jiangsu Huachuang Sino Medical Technology Co., Ltd. (Taizhou, China). They were randomly divided into 20 groups, with 6 mice in each group. They were raised under SPF conditions before the experiment. The vaccine dosage was administered to the mice at one-tenth of the human dosage. The human dosage required is 500 μL, so we set the injection dosage for the mice at 50 μL each time. PBS was used as the control group, and 50 μL of PBS was injected intramuscularly twice every 4 weeks. The positive control group and the experimental group were intramuscularly injected with 50 μL of vaccines containing different adjuvants in two doses. Two weeks after the last immunization, whole blood samples were collected. After centrifugation at 1000× *g* for 30 min, serum samples were obtained and stored at −80 °C before use. Meanwhile, spleen cells were collected. The spleen cells were ground and centrifuged at 400× *g* for 5 min. After treating the spleen cells with erythrocyte lysis solution and refrigerating them at −4 °C, the subsequent operations were continued [23]. The animal facility and animal management and use committee (IACAC-OD) of this experiment have been evaluated and approved by the International Laboratory Animal Evaluation and Accreditation Committee (Approval Number: HC24007-P001-01).

#### 2.2.2. gE Protein-Specific Antibodies Detection by Enzyme-Linked Immunosorbent Assay (ELISA)

The levels of gE protein-specific antibodies in serum samples collected from immunized mice were determined by ELISA. VZV gE dissolved in ELISA Coating Buffer (Solarbio, Beijing, China) was used to precoat 96-well microplates at a final concentration of 2.5 μg/mL. After overnight incubation at 4 °C, plates were washed three times with PBS containing 0.05% Tween 20 (PBST) and blocked for 1 h at 37 °C with 200 μL per well of blocking buffer (5% skimmed milk in PBST). Then the 96-well microplates were incubated with serially diluted sera (diluted from 1800 to 3,936,600) for another 1 h at 37 °C with 100 μL per well. After the plates were thoroughly washed using PBST, Goat anti-mouse IgG conjugated with horseradish peroxidase (HRP) was further added into the plates (1:5000, Southernbiotech, AL, USA) and allowed to incubate for another 1 h at 37 °C with 100 μL per well. Finally, 100 µL of TMB soluble (Beyotime, Changsha, China) was used for the color reaction, and the reaction was terminated via the addition of 50 µL Stop Solution for TMB Substrate (Beyotime, Changsha, China). The absorbance was measured at 450 nm using a Microplate reader (Multiskan, Shanghai, China). Using the OD value of the negative control group as the cutoff point at 2.1 times, the maximum dilution of serum with an OD value greater than the cutoff point represents the antibody titer of the positive group and the experimental group. The graph is plotted with the logarithm of the antibody titer values as the vertical coordinate values. The CV% of OD value of negative quality control samples was ≤20%, and the CV% of OD value of the sample was ≤20%.

#### 2.2.3. Flow Cytometry

Cellular immune responses were assessed by flow cytometry with intracellular cytokine staining for tumor necrosis factor alpha (TNF-α), interleukin-2 (IL-2), and IFN-γ, as previously described [24]. A total of 2 × 10^7^ splenocytes were plated in 96-well U-bottom plates and incubated overnight at 37 °C with 5% CO_2_. Then the splenocytes were stimulated with gE peptide pools at a final concentration of 2 μg/mL, DMSO (the same concentration), and PMA (Invitrogen, Carlsbad, CA, USA), incubated for 2 h at 37 °C with 5% CO_2_. The mixture was incubated for 4 h with Protein Transport Inhibitor Cocktail under the same conditions to block cytokine release. After washing with PBS, 100 μL of Zombie NIR^TM^ (Biolegend, San Diego, CA, USA) was added to each EP tube, and the tubes were incubated for 20 min. The cells were then stained using FITC anti-mouse CD3 antibodies (eBioscience, San Diego, CA, USA), Super Bright 600 anti-mouse CD4 antibodies (Bioscience, North Brunswick, NJ, USA), and PerCP/Cyanine5.5 anti-mouse CD8 antibodies (Biolegend, CA, USA) for 20 min. After washing with staining buffer, cells were then fixed with IC Fixation Buffer for 30 min and stained intracellularly with eFluor 450 anti-mouse IFN-γ antibodies (eBioscience, CA, USA), APC anti-mouse TNF-α antibodies (eBioscience, CA, USA), and PE anti-mouse IL-2 antibodies (Invitrogen, CA, USA) in the dark at room temperature for 1 h. After staining, the cells were gated (forward and side scatter, FSC/SSC), and samples with 100,000 cells were analyzed with a Cytek flow cytometer.

#### 2.2.4. Enzyme-Linked Immuno-Spot (ELISpot) Assay

Splenocytes (2 × 10^5^ cells/well) of immunized mice were seeded in 96-well plates for further analysis with enzyme-linked immunospot (ELISpot) assay kits (MabTech, Stockholm, Sweden, catalog number 3441-4APW-10 for IL-2 and 3321-4AST-10 for IFN-γ) according to the manufacturer’s protocol [25]. The protein gE at a final concentration of 2 μg/mL was used to stimulate gE-specific T cell responses, and the same volume of DMSO (Sigma, St. Louis, MO, USA) was used as a control group. The ConA (Invitrogen, CA, USA) was used as a positive control; IL-2 concentration was 1 µg/mL, IFN-γ concentration was 2 µg/mL. Spots were counted with an ELISpot reader system (Autoimmun Diagnostika GmbH, Strassberg, Germany) [26].

### 2.3. Rats Study

SPF Sprague Dawley rats (SD rats) were provided by Beijing Vital River Laboratory Animal Technology Co., Ltd. (Beijing, China). After isolation and quarantine, the animals were raised by Guoke Saifu Hebei Pharmaceutical Technology Co., Ltd. (Langfang, China). A total of 160 rats were randomly divided into a negative control group, an adjuvant control group, a candidate low dose group, and a candidate high dose group according to body weight, with 20 rats in each group of each sex. The body weight of individuals of the same sex before grouping was within the range of ±20% of the average body weight. The low-dose group received 0.25 mL of the SWE/CPG combined adjuvant vaccine combination per rat per injection, while the high-dose group received 0.5 mL. The negative control group and the adjuvant control group were injected with 0.5 mL of the negative control product (sodium chloride injection) or the adjuvant control product per rat, respectively. Muscle injections were administered to rats at the age of 6–7 weeks. During the injection, the weight of female rats ranged from 178.30 to 242.06 g, and that of male rats ranged from 240.34 to 307.33 g. The drug was given biweekly for 4 weeks, 3 times (D1, D15, D29), and then the drug was withdrawn for 4 weeks (D58). The day of the first dose was defined as day 1 of the trial (D1). The intramuscular immunization of rats, sample collection, establishment, and validation of immunogenicity test methods, and sample determination were performed by Guoke Saifu Hebei Pharmaceutical Technology Co., Ltd. The collection time of anti-drug antibody samples was before the first injection, D15 (before injection), D29 (before injection), D43 (2 weeks after the last injection), and D58 (end of the recovery period). The rat serum samples were obtained, and the titers of rat gE-specific immunoglobulin IgG antibodies were detected by ELISA. The serum anti-gE protein antibodies (ADA) were detected at D15 (2 weeks after the first injection), D32 (3 days after the last injection), D43 (2 weeks after the end of recovery), and D58 (end of the recovery period). The ELISpot method was used to detect the levels of specific IFN-γ induced by the peptide library derived from the gE protein in the spleen after stimulation by the peptide library at D15, D32, D43, and D58 after the injection. The contents and procedures related to animal experiments involved in this experiment comply with the relevant laws and regulations on the use and management of laboratory animals and the relevant regulations of the Institutional Animal Care and Use Committee (IACUC) of this institution. The animal number, experimental design, and animal handling have been approved by the IACUC of this institution, with the approval number: IACUC-2024-766.

### 2.4. Rhesus Monkeys Study

The common-grade rhesus monkeys were provided by Guangzhou Xiangguan Biotechnology Co., Ltd. (Guangzhou, China). After quarantine, they were raised by Guoke Saifu Hebei Pharmaceutical Technology Co., Ltd. (Langfang, China). Twenty-four rhesus monkeys were randomly divided into a negative control group, an adjuvant control group, a low-dose group, and a high-dose group based on their body weight, with a total of 4 groups, each containing 6 monkeys, with an equal number of males and females. The low-dose group received 1 dose of the SWE/CPG combined adjuvant vaccine per monkey, while the high-dose group received 2 doses per monkey. The negative control group and the adjuvant control group were given 1 mL per monkey of sodium chloride injection or the MF59/CPG combined adjuvant vaccine mixture by muscle injection. The rhesus monkeys were injected with a quadriceps muscle injection at the age of 3–5 years. During the injection, the weight range of female rhesus monkeys was 3.2–5.8 kg, and that of male rhesus monkeys was 3.1–4.4 kg. The injections were administered once every two weeks for 4 weeks (a total of 3 times, on D1, D15, and D29), followed by a 4-week recovery period. The first injection day was defined as the first day of the experiment (D1). The muscle immunization, sample collection, immunogenicity test methods, validation, and sample determination of the rhesus monkeys were all carried out by Guoke Saifu Hebei Pharmaceutical Technology Co., Ltd. Anti-gE antibodies in the serum were detected before the first injection, on D15, D29, D32, and D58; specific T cells in the peripheral blood were detected using ELISpot on D15, D29, D32, and D58. Guoke Saifu Hebei Pharmaceutical Technology Co., Ltd. was entrusted to conduct serum antibody detection for anti-gE antibodies at the time of the first injection, on D15 (before the injection), D29 (before the injection), D32, and D58; ELISpot was used to detect specific T cells (IFN-γ, IL-2) in peripheral blood at D15 (before the injection), D29 (before the injection), D32, and D58. All the contents and procedures related to the animal experiments in this trial followed the relevant laws and regulations for the use and management of experimental animals and the relevant regulations of the Institutional Animal Care and Use Committee (IACUC) of this institution. The animal number, trial design, and animal handling of this trial have all been approved by the IACUC of this institution, with approval numbers: IACUC-2024-791, IACUC-2025-164.

### 2.5. Statistical Analysis

GraphPad Prism 10.0 (GraphPad Software, San Diego, CA, USA) was used for data cleaning, analysis, and plotting. Data were expressed as mean ± standard deviation (mean ± SD). One-way or two-way ANOVA was used to compare the differences between groups, and Tukey’s multiple comparison test was used to compare the differences between the means of each group and the means of other groups. Asterisks represent the Pvalue classification: * *p* < 0.05; ** *p* < 0.01, *** *p* < 0.001, **** *p* < 0.0001.

## 3. Results

### 3.1. Comparative Analysis of the Effects of Different Adjuvant Doses on the Immune Level of Mice

To evaluate the effect of different adjuvant formulations on the humoral immune response in mice, we measured the ge-specific IgG titer in mouse serum by ELISA 2 weeks after the last immunization. We found that the expression level of gE antigen-specific IgG was significantly higher in the experimental groups with either the single adjuvant or the combined adjuvant than in the negative control group, which induced a favorable humoral immune response (Figure 1A,B), but the expression level of gE antigen-specific IgG was lower than that in the positive control group.

Spleen cells were collected 4 weeks after the last immunization. CD4^+^ T cells were stimulated with mixed peptides in vitro. Flow cytometry showed that different combinations of adjuvants could induce CD4^+^ T cells to secrete cytokines. In the antigen-vaccine combination-mediated CD4^+^ T lymphocyte secretion of IFN-γ, TNF-α, and IL-2 by flow cytometry, the combination of gE and SWE adjuvant could effectively promote Th1 cell response and enhance the secretion of cytokines at the level of cellular immune response, while the immunogenicity of the other two adjuvants was relatively low. At the same gE antigen concentration, adding 25 μL SWE (S4) could effectively improve the cytokine levels, and the cellular immune level of mice was similar to that of the positive control group, with no significant difference (*p* < 0.05) (Figure 1C). Comparison of immune responses induced by single-adjuvant vaccines and positive controls revealed slightly lower but not significantly different cytokine levels induced by all combinations of single-adjuvants compared with positive controls, possibly because single adjuvants were not sufficient to enhance the ability of gE antigen to induce strong immune responses. “With the addition of CpG1018 adjuvant to a single adjuvant, we found that the prepared recombinant herpes zoster vaccine composition consisting of aluminum hydroxide and CpG1018 or SWE adjuvant and CpG1018 could induce higher levels of cytokines, especially IFN-γ production by T cells.” To determine the final adjuvant combination, we performed a dose–response relationship study of SWE/aluminum hydroxide and adjuvant CpG1018. We kept the antigen content constant, and at SWE of 25 μL, 12.5 μL, or aluminum hydroxide of 50 μg, the immune responses of the mice in the experimental groups were evaluated after adding 2.5 μg, 5 μg, and 10 μg CpG1018, respectively. Using an ELISA, it was found that all nine compound adjuvant experimental groups could induce good humoral immune responses (Figure 1D), but the expression level of gE antigen-specific IgG was lower than that of the positive control group. In the cell-mediated immune response, the addition of CpG1018 to the adjuvant formulation was effective in inducing certain levels of IFN-γ, TNF-α, and IL-2. When the content of SWE was constant, the secretion of cytokines increased with the increase of CpG1018 content in each group. When the content of the aluminum hydroxide adjuvant was constant, the high dose of CpG adjuvant weakened the cellular immunity. When the content of SWE was 25 μL and the content of CpG1018 was 10 μg, the levels of cytokines induced by SWE were significantly stronger than those of other experimental groups (*p* < 0.05).

In addition, IFN-γ and IL-2 secreted by CD4^+^ T lymphocytes were detected by enzyme-linked immunospot assay for repeated verification. Similarly, in the single adjuvant group, the SWE adjuvant was more effective than the other adjuvants (Figure 1E). When SWE was 25 μL, the cellular immune level of mice reached the peak, which was lower than that of the positive control group, but there was no significant difference. When the content of SWE was constant, the cytokine content increased with the increase of CpG1018 (Figure 1F). When the content of CpG1018 was 10 μg, the cytokine content was the highest. When the content of aluminum hydroxide was constant, the high dose of CpG adjuvant weakened the cellular immunity, which was consistent with the results obtained by flow cytometry.

In previous pre-experiments, we found that the adsorption rate of aluminum hydroxide and gE antigens decreased over time, and the products became unstable. In summary, we selected the ge/SWE/CpG1018 composite adjuvant combination, which produced higher humoral and cellular immune responses, as the best adjuvant combination for the recombinant herpes zoster vaccine.

### 3.2. The Immunogenicity of the gE/SWE CpG1018 Vaccine Adjuvant Combination Was Tested in SD Rats

The immunogenicity of a recombinant herpes zoster vaccine (Group C8) was evaluated in rats administered at two different dose levels. We detected serum anti-gE protein antibody on D15, D29, D43, and D58 after the first administration, and found that the positive rate of serum anti-gE protein antibody was 100% in the rats injected with low and high doses of vaccine after administration. On day 15, the rats injected with low and high doses of vaccine were all detected positive for anti-gE protein antibody for the first time (Figure 2B). In addition, with the increase of immunization times, the antibody titer of gE protein in rats continued to increase, and it still showed a continuous increasing trend or no significant change until 2 weeks after the last administration (Figure 2A). After 4 weeks of recovery, it was found that the antibody titer of the low-dose group and the high-dose group remained high. During the dosing period, the antibody level in the high-dose group was slightly higher than that in the low-dose group, and there was no significant difference in antibody level between the groups after 4 weeks of recovery.

Furthermore, we used the ELISpot method to detect the specific levels of IFN-γ in the rat spleens at D15, D32, D43, and D58 after stimulation by the gE protein-derived peptide pool. The results showed that under the stimulation of the peptide pool, the negative control group and the adjuvant control group had negative test results. In each group of animals injected with low and high doses of IFN-γ, the specific T cell response was significantly enhanced. The specific T cell response in the low and high dose groups continued to increase at D32 and remained at a relatively high level at D43. After 4 weeks, a higher immune response was still detected (Figure 2B). In conclusion, these data indicate that the combined adjuvant vaccine group can effectively stimulate the humoral and cellular immunity of SD rats, confirming its strong immunogenicity potential.

### 3.3. The Immunogenicity of the gE/SWE CpG1018 Vaccine Adjuvant Combination Was Tested in Rhesus Monkeys

In addition to immunogenicity data in two rodent models, we evaluated the immunogenicity of C8 at two different dose levels in rhesus monkeys to facilitate later clinical translation. After administration, the first positive time of serum in rats injected with high and low doses of the C8 group was D15, and the positive rate reached 100% before the last administration (Figure 3B). As shown in Figure 3A, the antibody titer of the animals increased with the number of immunizations, and the antibody titer increased significantly at D29. Three days after the last dose (D32), the antibody titer of most animals showed a slight decrease or no significant change, and the antibody titer remained at a high level after 4 weeks of recovery.

In addition, under the stimulation of the peptide library, the results of the negative control group and the adjuvant control group were negative, and the specific T cells secreted IFN-γ and IL-2 in the experimental group injected with recombinant herpes zoster vaccine (Figure 3B,C). The specific T cell immune response reached the peak at D29, and the specific T cell immune response was still detected at the end of the recovery period. Moreover, the cellular immune response induced by the recombinant herpes zoster vaccine was inversely related to the dose administered. These results demonstrated that the combined adjuvant vaccine could effectively stimulate humoral and cellular immunity in rhesus monkeys.

## 4. Discussion

Herpes zoster is caused by reactivation of varicella zoster virus (VZV) that is latent in patients after self-healing of varicella [27,28]. VZV inhibits the expression of MHC-I and MHC-II molecules and activates autophagy of T lymphocytes through immune regulation, making it difficult for the target cells infected with VZV to be recognized by the immune system, and VZV will be reactivated. The reactivation process of VZV is closely related to the recognition of the virus by pattern recognition receptors, the interaction of T lymphocyte subsets, and the release of cytokines. The decrease of immune function is the key factor for VZV reactivation to develop into HZ. The expression and immune function of CD4^+^ T and CD8^+^ T cells play different roles in the occurrence and development of HZ. CD4^+^ T cells play an indispensable role in the acute and subclinical stages of HZ [29,30]. During reactivation of latent VZV, memory CD4^+^ T cells can rapidly recognize the virus and induce the release of various cytokines that facilitate immune clearance. The immune response caused by CD8^+^ T cells also plays a crucial role in the pathogenesis of HZ. CD8^+^ T cells can recognize more kinds of VZV-infected cells and exert a killing effect by secreting cytokines [31,32,33]. However, the number of B cells and T cells in lymphocytes decreases significantly with age, so it is important to generate a stable and high level of cellular immune response after vaccination in the elderly population to improve the effectiveness of vaccination in the elderly. Based on these previous studies, it can be seen that the immune response mechanisms caused by the invasion of the VZV in the human body are mostly related to cellular immunity. Therefore, this study focused on detecting the level of cellular immune response produced after administering the drug to animals. Only the level of humoral immunity in the animals’ bodies after administration was detected by combining antibody tests, and no further in-depth detection of humoral immune responses, such as neutralizing antibody tests, was conducted. This is also one of the limitations of this study.

The recombinant herpes zoster vaccine used in this study adopts recombinant DNA technology, and the truncated coding sequence of varicella zoster virus gE is injected into CHO cells to express varicella zoster virus gE, which can be produced on a large scale, at low cost, and with high efficiency through genetic engineering technology. The composition of recombinant protein vaccines is simple, which can reduce or eliminate harmful reactants such as pyrogens, allergens, and immunosuppressants that are difficult to avoid in traditional attenuated or inactivated vaccines. Since the recombinant protein vaccine cannot replicate in vivo, it has no pathogenic risk to the host and is safe and stable. All components of the adjuvant, including CpG stimulators and MF59, have been approved by the FDA for use in vaccines for humans and have shown a good safety profile [12,34]. This study focuses on selecting efficient and low-cost adjuvant formulas by adjusting the formulation of adjuvant. In terms of the production technology of recombinant proteins, more attention is paid to the safety and large-scale production of recombinant proteins, and there is not much research on the direction of reducing the production cost of recombinant proteins. In addition to the experiments described in this article, we also conducted safety pharmacological, local irritation, and immunological studies on rats and rhesus monkeys. The animals that received the gE/SWE/CpG1018 and adjuvant vaccine combination showed no abnormalities in clinical observations, body weight, body temperature, food intake, etc. During the experiments, no animal deaths or other obvious toxic reactions occurred. In the immunogenicity test of rhesus monkeys, compared with the low-dose group, the high-dose group of rhesus monkeys had less secretion of cytokines. This might be because one dose of the E/SWE/CpG1018 and adjuvant vaccine combination had already triggered a relatively high immune response in rhesus monkeys, and when the dose was increased to two doses, the immune response triggered by the combination of adjuvant and antigen was lower than that of one dose. However, all toxicological experiments were conducted in animal models, and further verification is needed in larger-scale primate studies to strengthen the evidence base. Besides its high safety advantages, the formulation of gE/SWE/CpG1018 and adjuvant vaccine combination contains all these components that can be synthesized in large quantities by artificial means and have lower costs. This indicates that it is possible to produce a more cost-effective varicella zoster vaccine. These results showed that this vaccine candidate could be used as a new vaccine for the prevention of varicella zoster and has good development prospects.

The low herpes zoster vaccination intention in China is mainly due to the low awareness of vaccination among the target population and the high cost of the herpes zoster vaccine in the market. Under the influence of the social atmosphere that pays more and more attention to the quality of life and health, as well as the reality of the lack of effective therapeutic drugs for HZ, vaccination has become an inevitable choice for some middle-aged and elderly people to prevent herpes zoster, and the expected market prospect is broad. Future research and development of recombinant herpes zoster vaccines should continue to improve the efficacy of the vaccine to ensure that it can provide long-term and effective protection for recipients, improve the immunogenicity and safety of the vaccine, and enhance the efficacy of the vaccine. With the continuous progress of technology and the continuous expansion of the market, the recombinant herpes zoster vaccine will play a more important role in the prevention of herpes zoster.

## 5. Conclusions

In this study, we identified a new vaccine formulation for the prevention of varicella zoster with an optimal antigen-adjuvant ratio of 5 μg gE, 25 μL, SWE and 10 μg CpG1018. The gE/SWE/CpG1018 combined with adjuvant vaccine showed good immunogenicity in C57BL/6 mice, SD rats, and rhesus monkeys. This formula has low cost and high efficiency and has good promotion prospects.

## Figures and Tables

**Figure 1 vaccines-13-01124-f001:**
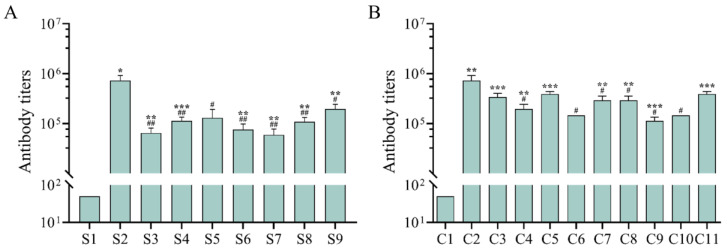
The levels of humoral and cellular immune responses induced in mice by different adjuvants. The data in the figure are presented as the mean and standard deviation (SD). (**A**,**B**) The specific IgG titers of VZV-gE antigen in the serum of mice 28 days after the second immunization were detected by enzyme-linked immunosorbent assay. (**A**) shows the single adjuvant combination group; (**B**) shows the composite adjuvant combination group. ***: *p* < 0.001, **/##: *p* < 0.01, */#: *p* < 0.05 indicates statistical differences between the two groups, ‘*’ indicates statistical differences between the experimental group and the negative group, ‘#’ indicates statistical differences between the experimental group and the positive group. (**C**,**D**) After peptide library stimulation, the levels of IFN-γ, TNF-α, and IL-2 cytokines produced by spleen cells of mice 28 days after the second immunization were detected by flow cytometry. (**C**) shows the single adjuvant group; (**D**) shows the composite adjuvant group. ###/bbb: *p* < 0.001, **/##/bb: *p* < 0.01, */#/a/b: *p* < 0.05 indicate statistical differences between the experimental group and the positive group, ‘*’ indicates statistical differences between the experimental group and the positive group of cytokines IFN-γ, ‘#’ indicates TNF-α, ‘a’ indicates IL-2, ‘b’ indicates the total number of cytokines. (**E**–**G**) The levels of gE-specific cytokines obtained by spleen cells of mice were detected by ELISpot. (**E**) shows the single adjuvant group, (**F**) shows the composite adjuvant group, and (**G**) shows the PBS control group, the positive stimulation group, the C2 group, and C8 group. ***/###: *p* < 0.001, **: *p* < 0.01, *: *p* < 0.05 indicate statistical differences between the experimental group and the positive group, ‘*’ indicates statistical differences between the experimental group and the positive group of cytokines IFN-γ, and ‘a’ indicates the total number of cytokines.

**Figure 2 vaccines-13-01124-f002:**
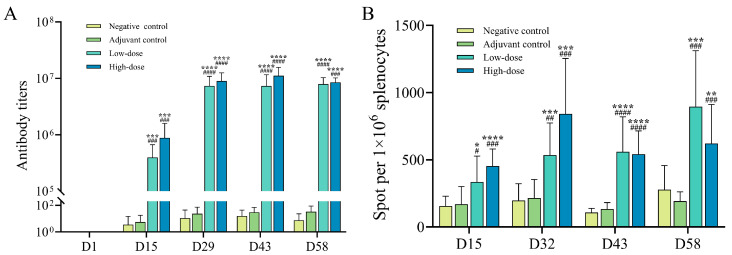
(**A**) ELISA test results showing the titers of anti-gE protein antibodies in the serum of each group of SD rats. (**B**) ELISpot test results indicating the number of IFN-γ spots produced by spleen cells of each group of SD rats after stimulation with peptide libraries at different collection points. ****/####: *p* < 0.0001, ***/###: *p* < 0.001, **/##: *p* < 0.01, */#: *p* < 0.05 indicate statistical differences between the two groups, ’*’ indicates a statistical difference between the low-dose group and the negative control group, and ’#’ indicates a statistical difference between the high-dose group and the negative control group.

**Figure 3 vaccines-13-01124-f003:**
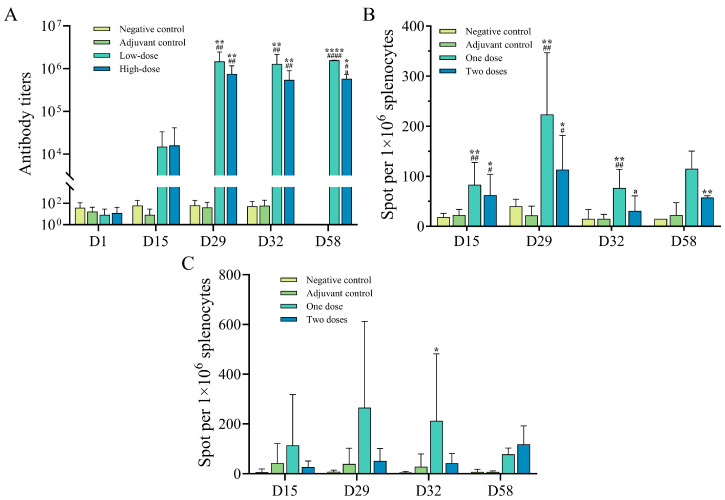
(**A**) ELISA test results for the antibody titers against gE protein in the serum of each group of rhesus monkeys at different collection time points. (**B**,**C**) ELISpot test results for the number of IL-2 (**B**) and IFN-γ (**C**) spots produced by the spleen cells of each group of rhesus monkeys after stimulation with the peptide library. ****/####: *p* < 0.0001, **/##: *p* < 0.01, */#: *p* < 0.05 indicate statistical differences between the two groups, ’*’ indicates a statistical difference between the low-dose group and the negative control group, ‘#’ indicates a statistical difference between the high-dose group and the negative control group, and ‘a’ indicates a statistical difference between the high-dose group and the low-dose group.

**Table 1 vaccines-13-01124-t001:** Single adjuvant immunization group.

Group	Vaccine Composition (50 μL System)
S1	PBS
S2	gE (5 µg) + AS01_B_ (50 µL)
S3	gE (5 µg) + aluminum hydroxide (50 µg)
S4	gE (5 µg) + SWE (25 µL)
S5	gE (5 µg) + SWE (45 µL)
S6	gE (5 µg) + CpG1018 (2.5 µg)
S7	gE (5 µg) + CpG1018 (10 µg)
S8	gE (5 µg) + CpG1018 (40 µg)
S9	gE (5 µg) + CpG1018 (200 µg)

**Table 2 vaccines-13-01124-t002:** Compound adjuvant immune group.

Group	Vaccine Composition (50 μL System)
C1	PBS
C2	gE (5 µg) + AS01_B_ (50 µL)
C3	gE (5 µg) + aluminum hydroxide (50 µg) + CpG1018 (2.5 µg)
C4	gE (5 µg) + aluminum hydroxide (50 µg) + CpG1018 (5 µg)
C5	gE (5 µg) + aluminum hydroxide (50 µg) + CpG1018 (10 µg)
C6	gE (5 µg) + SWE (25 µL) + CpG1018 (2.5 µg)
C7	gE (5 µg) + SWE (25 µL) + CpG1018 (5 µg)
C8	gE (5 µg) + SWE (25 µL) + CpG1018 (10 µg)
C9	gE (5 µg) + SWE (12.5 µL) + CpG1018 (2.5 µg)
C10	gE (5 µg) + SWE (12.5 µL) + CpG1018 (5 µg)
C11	gE (5 µg) + SWE (12.5 µL) + CpG1018 (10 µg)

## Data Availability

Data will be made available upon request.

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
