# Peer review of "Effect of Different Adjuvants on the Immunogenicity of a Recombinant Herpes Zoster Vaccine in Mice, Rats and Non-Human Primates"

_vaccines, 2025, doi:10.3390/vaccines13111124_

Round 1
Reviewer 1 Report
Comments and Suggestions for Authors
These are my comments:
Line 114, "Chinese hamster ovary(CHO) cell-expressed VZV gE extracellular domain protein", how were these cells obtained? Are they commercial?
Line 142, "1/10 of the human dosage", what is the composition of the human dose?
Line 159, "candidate low and high-dose groups", what are the low and high dose groups?
Line 160, "0.5 doses of MF59/CPG", what is the composition of this dose?
Line 186, "low and high doses", specify this?
Line 249, "* indicates P < 0.05, indicating a significant difference, * indicates P < 0.01", same single asterisk notation?
Line 255, "ge-specific"? gE?
Line 280, "25μL, 12.5μL, or aluminum hydroxide of 50μg", separate units from dimensions i.e., 25 μL. Do this throughout the document,
Figure 1, it seems to me that C3 is as good as C8, comment on this.
Line 324, "**: p < 0.001, **: p < 0.01"?
Figure 2, why the controls show positive rate?
Figure 2, there are two graphs of positive rate (B and C), only A and B are described in the figure's caption.
Line 368, "As shown in Figure 2A", is it Figure 3A?
Line 427-428, "no abnormalities were observed in any of the indicators", what indicators?
Line 415-417, "The recombinant herpes zoster vaccine used in this study adopts recombinant DNA technology, and the truncated coding sequence of varicella-zoster virus gE is injected into CHO cells to express varicella-zoster virus gE", this was poorly described in the document.
Line 433, "to produce a more cost-effective herpes zoster vaccine", using E. coli to produce the recombinant protein would be way cheaper than using CHO cells, comment on this.
Discussion, comment on the limitations of the study presented. Are the specific antibodies neutralizing? A challenge must be performed also.
Author Response
Comments 1: [Line 280, "25μL, 12.5μL, or aluminum hydroxide of 50μg", separate units from dimensions i.e., 25 μL. Do this throughout the document
Line 249, "* indicates P < 0.05, indicating a significant difference, * indicates P < 0.01", same single asterisk notation?
Line 324, "**: p < 0.001, **: p < 0.01"?
Line 368, "As shown in Figure 2A", is it Figure 3A?]
Response 1: [Thank you for your careful reminder. The problem here is an input error.]We agree with this comment, and all the revisions have been made throughout the entire text.
Comments 2: [Chinese hamster ovary(CHO) cell-expressed VZV gE extracellular domain protein", how were these cells obtained? Are they commercial?]
Response 2: [The plasmid containing the target gene (amino acids 31 to 538 of VZV-gE protein) was transfected into CHO cells. After obtaining the monoclonal cell line, cell culture and protein expression, purification were carried out. Finally, the gE protein was obtained.] Thank you for pointing this out. I/We agree with this comment.
Comments 3: [Line 142, "1/10 of the human dosage", what is the composition of the human dose?]
Response 3: [The human dosage required is 500 μL, so we set the injection dosage for the mice at 50 μL each time.] Thank you for pointing this out. We agree with this comment.We have already updated this part in line 161 of the main text.
Comments 4: [Line 159, "candidate low and high-dose groups", what are the low and high dose groups?]
Response 4: [The candidate low and high-dose groups were intramuscularly injected with 0.5 doses of MF59/CPG composite adjuvant vaccine combinations at 0.25 and 0.5 mL per rat per injection, respectively, and the negative control group and the adjuvant control group were intramuscularly injected with negative control products(sodium chloride injection) or adjuvant control products at 0.5 mL per rat volume.]This is located on line 228 of the main text.
Comments 5: [Line 160, "0.5 doses of MF59/CPG", what is the composition of this dose?]
Response 5: [This is the candidate adjuvant combination selected by our research.]
Comments 6: [Line 186, "low and high doses", specify this]
Response 6: [A total of 24 rhesus monkeys were randomly divided by body weight into a negative control group, an adjuvant control group, and three low-dose and high-dose groups. The low-dose group was given 1 dose of the MF59/CPG combined adjuvant vaccine per monkey, while the high-dose group was given 2 doses of the MF59/CPG combined adjuvant vaccine per monkey.] Thank you for pointing this out. We agree with this comment.We have already updated this part in line 255 of the main text.
Comments 7: [Line 255, "ge-specific"? gE? gE?]
Response 7: [This refers to the specific antibodies against the varicella-zoster virus.]
Comments 8: [Figure 1, it seems to me that C3 is as good as C8, comment on this.]
Response 8: [The results of humoral immunity are only used as a secondary screening criterion. Our main focus is on the data of cellular immunity.]
Comments 9: [Line 427-428, "no abnormalities were observed in any of the indicators", what indicators?]
Response 9: [Here are the indicators such as clinical observations, body weight, and food intake.]
Comments 10: [Line 415-417, "The recombinant herpes zoster vaccine used in this study adopts recombinant DNA technology, and the truncated coding sequence of varicella-zoster virus gE is injected into CHO cells to express varicella-zoster virus gE", this was poorly described in the document.
Line 433, "to produce a more cost-effective herpes zoster vaccine", using E. coli to produce the recombinant protein would be way cheaper than using CHO cells, comment on this.
Discussion, comment on the limitations of the study presented. Are the specific antibodies neutralizing? A challenge must be performed also.]
Response 10: We agree with this comment.We will update this section and the discussion content in the subsequent manuscript. Thank you for your valuable feedback.
The specific content of the manuscript will be submitted after unified revision. Thank you for your valuable suggestions.
Reviewer 2 Report
Comments and Suggestions for Authors
The authors report on preclinical studies on the development of a new gE-based vaccine with different combinations of adjuvants for the prevention of shingles. The authors suggest that the licensed Shingrix vaccine, based on the gE protein with the AS01 combination adjuvant, is effective, but is expensive and has limited availability because of the inclusion of QS-21. They evaluated a squalene in water emulsion combined with the CpG1018 adjuvant in mice, rats and rhesus macaques. The results suggest that the emulsified CpG1018 adjuvant is a potential alternative to AS01.
A strength of the study is the use of three different animal models, inclusion of AS01 as a comparator and positive control, and assessment of both the anti-gE antibody and CD4 T cell responses. However, the presentation and analysis of the data is flawed making it difficult to evaluate the results. In addition, the description of the methods is incomplete.
Major comments:
- The authors state that “Statistical analysis of the differences between groups was conducted using the t-test”. This is not appropriate for experiments with more than two groups. All data presented in Figures 1-3 should be analyzed by ANOVA followed by an appropriate test to compare groups with multiple comparison correction for type I errors (e.g., Tukey’s, Bonferroni). Antibody titers should be defined (see comment 15.) and log-transformed titers should be used for the statistical analysis.
- Figures 2B and C and 3 B-D. The authors should explain what they mean by positive rate. Why did the positive rate increase in the vehicle control and adjuvant control groups in Figure 2B? They should also present the number of IFN-g and IL-2 positive cells in Figure 2C and 3C and D in the more conventional form of histograms either in addition to or instead of the “positive rate”.
- Line 116. To the best of my knowledge, MF59 is a proprietary adjuvant and not manufactured or sold by Seppic. The authors should describe the adjuvant that they purchased from Seppic and may refer to it as “MF59-like” or a “squalene oil-in-water emulsion” (e.g. abbreviated SWE).
- Sections 2.3 – 2.5. The methods are only described for mice. Please include a description of these assays for rats and rhesus monkeys.
Other comments:
- Title: “Animal Bodies” is awkward. Suggest to change to “Effect of different adjuvants on the immunogenicity of a recombinant Herpes Zoster vaccine in mice, rats and non-human primates.”
- Line 33, 34. Delete this sentence. The “immunogenicity” is repetitive with the previous sentence, and the authors did not really determine the duration of the immune response (see also comments 16 and 18.).
- Line 44,45. Change to “…weakened immune systems due to old age or immunosuppression….”
- Line 50. Define PHN.
- Lines 87-89. “….protecting the host from pathogens. Enhance the secretion of cytokines, and then enhance the specific immune response after vaccination [16]. This is not a grammatical sentence. Also, CpG by itself does not protect the host from pathogens.
- Line 127. Identify the source of AS01 and whether it is AS01b or AS01e.
- Line 132-133. Identify the source of the aluminum hydroxide adjuvant.
- Line 160. Animals do not have genders. Replace by “sex” or “male and female”.
- Line 160-161. “with 0.5 doses of MF59/CPG composite adjuvant vaccine combinations at 0.25 and 0.5 mL per rat per injection”. This is unclear. A 0.5 dose would be 0.25 mL.
- Line 189. “….vaccine at a volume of 0.5 or 1 mL per monkey.” What was the rationale for administering twice the human dose to the monkeys?
- Section 2.3. (ELISA). Please add how the antibody titers were determined.
- Line 354. “…,with some immune persistence.” Please remove this. The authors only extended the observations to 58 days which does not provide any indication of the duration of the immune response.
- Line 375. “….no significant gender difference (Figure 3C-D).” Animals do not have genders. Replace by “sex difference”. Figures 3C and 3D do not show the differences between male and female monkeys.
- Line 381. Remove “and had a certain persistence” (see also comment 6. and 16.).
- Given the similar results of aluminum hydroxide adjuvant (AH) with CpG and the MF59-like adjuvant with CpG, the authors should discuss why they selected the emulsion, especially since an AH/CpG combination adjuvant is already used in licensed COVID-19 vaccines.
- Line 400 “CD4+ T cells play an indispensable role in the acute and subclinical stages of HZ.” Please provide one or more references.
- Line 420. Replace “pyogenes” with “pyrogens”.
- Line 425-429. Please provide a reference for this if published or add “(unpublished observations)”.
- Most of the references lack the year of publication and some do not have page numbers.
Can be improved.
Author Response
Comments 1: [The authors state that “Statistical analysis of the differences between groups was conducted using the t-test”. This is not appropriate for experiments with more than two groups. All data presented in Figures 1-3 should be analyzed by ANOVA followed by an appropriate test to compare groups with multiple comparison correction for type I errors (e.g., Tukey’s, Bonferroni). Antibody titers should be defined (see comment 15.) and log-transformed titers should be used for the statistical analysis.]
Response 1:[ Agree.We have revised the data analysis content in lines 280- 285.]
Comments 2: [ Figures 2B and C and 3 B-D. The authors should explain what they mean by positive rate. Why did the positive rate increase in the vehicle control and adjuvant control groups in Figure 2B? They should also present the number of IFN-g and IL-2 positive cells in Figure 2C and 3C and D in the more conventional form of histograms either in addition to or instead of the “positive rate”.]
Response 2:[Agreed. We have replaced the relevant pictures and revised the article content. This has been done at line 224 of the main text.]
Comments 3: [Line 116. To the best of my knowledge, MF59 is a proprietary adjuvant and not manufactured or sold by Seppic. The authors should describe the adjuvant that they purchased from Seppic and may refer to it as “MF59-like” or a “squalene oil-in-water emulsion” (e.g. abbreviated SWE).]
Response 3:[ Agree.We have replaced the MF59 adjuvant produced by Seppic with the abbreviation SWE in the full text.]
Comments 4: [Sections 2.3 – 2.5. The methods are only described for mice. Please include a description of these assays for rats and rhesus monkeys.]
Response 4:[ The establishment, verification and sample measurement of the relevant experimental methods for rats and rhesus monkeys were carried out by Guoxuesai Fu Hebei Pharmaceutical Technology Co., Ltd. There is not much difference from the mouse experimental protocol, so it was not elaborated in the main text. If necessary, we can add it in the article.]
Comments 5: [Title: “Animal Bodies” is awkward. Suggest to change to “Effect of different adjuvants on the immunogenicity of a recombinant Herpes Zoster vaccine in mice, rats and non-human primates.”]
Response 5: [Agree. We have revised the title.]
Comments 6: [Line 33, 34. Delete this sentence. The “immunogenicity” is repetitive with the previous sentence, and the authors did not really determine the duration of the immune response (see also comments 16 and 18.).]
Response :[ Agree. We have removed these uncertainties at line 45.]
Comments 7: [Line 44,45. Change to “…weakened immune systems due to old age or immunosuppression….”]
Response 7:[ Agree.We have replaced this section on line 55.]
Comments 8: [Line 50. Define PHN.]
Response 8:[Agree.We have supplemented this part of the text at line 61.]
Comments 9: [Lines 87-89. “….protecting the host from pathogens. Enhance the secretion of cytokines, and then enhance the specific immune response after vaccination [16]. This is not a grammatical sentence. Also, CpG by itself does not protect the host from pathogens.]
Response 9:[Agree.We have supplemented this part of the text at line 99-103.]
Comments 10-11: [Line 127. Identify the source of AS01 and whether it is AS01b or AS01e.] [Line 132-133. Identify the source of the aluminum hydroxide adjuvant.]
Response 10-11:[ Agree. We have rewritten the administration method section, adding detailed information such as the sources of each adjuvant, starting from line 127 to line 151 of the article.]
Comments 12: [Line 160. Animals do not have genders. Replace by “sex” or “male and female”.]
Response 12:[ Agree. We have replaced this section on line 228.]
Comments 13: [Line 160-161. “with 0.5 doses of MF59/CPG composite adjuvant vaccine combinations at 0.25 and 0.5 mL per rat per injection”. This is unclear. A 0.5 dose would be 0.25 mL.]
Response 13:[Agree. We have made modifications to lines 1228-233 of this section and replaced it with Muscle injections were performed on the rats. The low-dose group received 0.25 ml of the SWE/CPG combined adjuvant vaccine combination per rat per injection, while the high-dose group received 0.5 ml. The negative control group and the adjuvant control group were respectively injected with 0.5 ml of the negative control product (sodium chloride injection) or the adjuvant control product per rat.]
Comments 14: [Line 189. “….vaccine at a volume of 0.5 or 1 mL per monkey.” What was the rationale for administering twice the human dose to the monkeys?]
Response 14:[A double dose can provide the maximum exposure to potential toxic effects in preclinical trials, enabling a more comprehensive assessment of the vaccine's safety.]
Comments 15: [Section 2.3. (ELISA). Please add how the antibody titers were determined.]
Response 15:[Agree. We have performed a supplement to the ELISA-related experimental protocol in lines 188-194.]
Comments 16: [Line 354. “…,with some immune persistence.” Please remove this. The authors only extended the observations to 58 days which does not provide any indication of the duration of the immune response.]
Response 16:[ Agree.We have removed the relevant content.]
Comments 17: [Line 375. “….no significant gender difference (Figure 3C-D).” Animals do not have genders. Replace by “sex difference”. Figures 3C and 3D do not show the differences between male and female monkeys.]
Response 17:[ Agree.We have removed the content not covered in the image.]
Comments18: [Line 381. Remove “and had a certain persistence” (see also comment 6. and 16.).]
Response 18:[ Agree.We have removed the relevant content.]
Comments 19: [Given the similar results of aluminum hydroxide adjuvant (AH) with CpG and the MF59-like adjuvant with CpG, the authors should discuss why they selected the emulsion, especially since an AH/CpG combination adjuvant is already used in licensed COVID-19 vaccines.]
Response 19: [ Agree. In our pre-experiment, as time went on, the adsorption rate of aluminum hydroxide and the gE antigen decreased, and the product became less stable.We have already included the conclusion for this part in the conclusion section.We have supplemented the conclusions of this section in lines 339 to 343.]
Comments 20: [Line 400 “CD4+ T cells play an indispensable role in the acute and subclinical stages of HZ.” Please provide one or more references.]
Response 20:[ Agree.We have added this section of references. The titles of the two references are as follows:‘Immune Responses to a Recombinant Glycoprotein E Herpes Zoster Vaccine in Adults Aged 50 Years or Older’, ‘ Role of regulatory T cells in patients with acute herpes zoster and relationship to postherpetic neuralgia’.]
Comments : [Line 420. Replace “pyogenes” with “pyrogens”.]
Response : [We completed the replacement at line 446 of the main text and marked it with red font.Thank you for pointing this out.]
Comments : [Line 425-429. Please provide a reference for this if published or add “(unpublished observations)”.Most of the references lack the year of publication and some do not have page numbers.]
Response : [This section presents the content of this experiment. We have provided additional relevant information in the methods section. Thank you for your feedback. We will make further revisions to the reference section.]
We are currently making further revisions to the manuscript based on your suggestions. Thank you very much for your valuable input, which has been of great help to us in completing the article.
Round 2
Reviewer 2 Report
Comments and Suggestions for Authors
Revision 1.
The authors have addressed some of my comments, however, several issues remain.
- Under Statistical analysis, the authors state “The data were analyzed using one-way analysis of variance (ANOVA), and statistical analysis was performed using the Student's t-test.” This is confusing and not correct. The authors used more than 2 experimental groups in their experiments, and Student’s t-test is therefore not acceptable. Instead, a one-way ANOVA should be performed and, if significant, a post-hoc analysis with correction for multiple comparisons should be performed (e.g. Bonferroni or Tukey’s). This applies to all data shown in Figure 1 A-F. For Figures 2 and 3 the authors should consider a two-way ANOVA as they have time and treatment as variables.
- The Abstract contains some misleading statements. The authors should make it clear in the abstract that the SWE and SWE/CpG were compared with AS01 and aluminum hydroxide (AH) alone or with CpG. The statements that “the formulation containing gE and SWE adjuvant induced the strongest immune response when using a single adjuvant” and “When a second adjuvant was added, the SWE/CpG1018 composite adjuvant performed the best” are not correct as AS01 and AH/CpG seem to perform as well as the SWE/CpG adjuvant.
- Line 133 “This experiment selected three adjuvants……”. At this place, the authors should also add AS01 as an adjuvant as it was used as a positive control.
- Line 143. “AS01 adjuvant from Influenza Vaccine (GSK, UK)” This is not correct information as there is no influenza vaccine with AS01 as adjuvant. The authors should also specify whether the AS01 is AS01b or AS01e.
- Response to comment 4. “The establishment, verification and sample measurement of the relevant experimental methods for rats and rhesus monkeys were carried out by Guoxuesai Fu Hebei Pharmaceutical Technology Co., Ltd. There is not much difference from the mouse experimental protocol, so it was not elaborated in the main text. If necessary, we can add it in the article”. The authors should add at least the strain and age of the rats and the age of the rhesus macaques.
- Response to comment 3 “We have replaced the MF59 adjuvant produced by Seppic with the abbreviation SWE in the full text”. The authors still refer to MF59 in the Introduction (line 80-81). They should also define SWE when it is first used in the Abstract.
- Response to comment 5. “Agree. We have revised the title.” However, the title in the revised manuscript was not changed.
- Response to comment 14. “A double dose can provide the maximum exposure to potential toxic effects in preclinical trials, enabling a more comprehensive assessment of the vaccine's safety.” The authors observed a lower IL-2 and to some extent IFN-gamma secretion in rhesus macaques immunized with the double dose compared with the single dose. Is this a toxicologic effect? The authors should discuss this.
- Response to comment 19. “ Agree. In our pre-experiment, as time went on, the adsorption rate of aluminum hydroxide and the gE antigen decreased, and the product became less stable.We have already included the conclusion for this part in the conclusion section.We have supplemented the conclusions of this section in lines 339 to 343.” The authors should provide more details – over what time period did the adsorption decline? What do they mean by less stable?
- Conclusions; 5. Discussion. It is awkward to have the Conclusions before the Discussion. I suggest to switch the order.
- References are still incomplete (missing page numbers), e.g., Refs 3, 14, 15, 19, 22, 24, 32, 34.
Can be improved.
Author Response
Comments 1 : [ Under Statistical analysis, the authors state “The data were analyzed using one-way analysis of variance (ANOVA), and statistical analysis was performed using the Student's t-test.” This is confusing and not correct. The authors used more than 2 experimental groups in their experiments, and Student’s t-test is therefore not acceptable. Instead, a one-way ANOVA should be performed and, if significant, a post-hoc analysis with correction for multiple comparisons should be performed (e.g. Bonferroni or Tukey’s). This applies to all data shown in Figure 1 A-F. For Figures 2 and 3 the authors should consider a two-way ANOVA as they have time and treatment as variables.]
Response 1 : [ Agree. We modified this section in line 285 of the paper.’Graphpad Prism 10.0 was used for data cleaning, analysis, and plotting. Data were expressed as mean± standard deviation (mean±SD). One-way or two-way ANOVA was used to compare the differences between groups, and Tukey's multiple comparison test was used to compare the differences between the means of each group and the means of other groups. Asterisks represent the Pvalue classification: *p<0.05; **p <0.01; ***p <0.001, ****p <0.0001.’]
Comments 2 : [ The Abstract contains some misleading statements. The authors should make it clear in the abstract that the SWE and SWE/CpG were compared with AS01 and aluminum hydroxide (AH) alone or with CpG. The statements that “the formulation containing gE and SWE adjuvant induced the strongest immune response when using a single adjuvant” and “When a second adjuvant was added, the SWE/CpG1018 composite adjuvant performed the best” are not correct as AS01 and AH/CpG seem to perform as well as the SWE/CpG adjuvant.]
Response 2 : [ Agree. We modified the contents of the abstract section.]
Comments 3-4 : [ Line 133 “This experiment selected three adjuvants……”. At this place, the authors should also add AS01 as an adjuvant as it was used as a positive control.]
[ Line 143. “AS01 adjuvant from Influenza Vaccine (GSK, UK)” This is not correct information as there is no influenza vaccine with AS01 as adjuvant. The authors should also specify whether the AS01 is AS01b or AS01e.]
Response 3-4 : [ Agree, we have updated the content of these two questions in lines 131 of the paper.]
Comments 5 : [ Response to comment 4. “The establishment, verification and sample measurement of the relevant experimental methods for rats and rhesus monkeys were carried out by Guoxuesai Fu Hebei Pharmaceutical Technology Co., Ltd. There is not much difference from the mouse experimental protocol, so it was not elaborated in the main text. If necessary, we can add it in the article”. The authors should add at least the strain and age of the rats and the age of the rhesus macaques.]
Response 5 : [ Agree. We have respectively added these pieces of information in the corresponding sections, which are located at line 224 and line 258.]
Comments 6 : [ Response to comment 3 “We have replaced the MF59 adjuvant produced by Seppic with the abbreviation SWE in the full text”. The authors still refer to MF59 in the Introduction (line 80-81). They should also define SWE when it is first used in the Abstract.]
Response 6 : [ Agree. We supplemented this section on line 86 of the paper.]
Comments 7 : [ Response to comment 5. “Agree. We have revised the title.” However, the title in the revised manuscript was not changed.]
Response 7 : [ Agree. Sorry, we have replaced the title.]
Comments 8 : [ Response to comment 14. “A double dose can provide the maximum exposure to potential toxic effects in preclinical trials, enabling a more comprehensive assessment of the vaccine's safety.” The authors observed a lower IL-2 and to some extent IFN-gamma secretion in rhesus macaques immunized with the double dose compared with the single dose. Is this a toxicologic effect? The authors should discuss this.]
Response 8 : [ Agree. We also discussed adding some additional content to this section of the discussion. We suspect that this is not caused by the toxic effect. In our remaining experimental data, such as general clinical indicators, detailed clinical indicators, and local stimulation observations, it was shown that in both dosage groups of the animals, no animals died or showed other obvious toxic reactions. We speculate that this phenomenon might be due to the fact that a single dose in rhesus monkeys triggered a relatively high immune response and maintained effective safety. When the dose was increased to 2 doses, the combination of adjuvant and antigen did not represent the optimal dose for triggering an immune response reaction. Therefore, the cellular immune data of 2 doses was lower than that of 1 dose.]
Comments 9 : [ Response to comment 19. “ Agree. In our pre-experiment, as time went on, the adsorption rate of aluminum hydroxide and the gE antigen decreased, and the product became less stable.We have already included the conclusion for this part in the conclusion section.We have supplemented the conclusions of this section in lines 339 to 343.” The authors should provide more details – over what time period did the adsorption decline? What do they mean by less stable?]
Response 9 : [ The adsorption rate of the preparation should generally be more than 90%. When combined with the aluminum hydroxide adjuvant, when developing liquid preparations, the adsorption rate usually decreases, and the stability indicators such as antigen content and immunogenicity will be affected.Below are some of our detection results for adsorption rates:]
|
Prescription dosage |
Adsorption rate |
||
|
|
5 min |
2 h |
4 h |
|
gE (5 μg) + Aluminum hydroxide (50 μg) + CpG (2.5 μg) |
77% |
71% |
59% |
|
gE (5 μg) + Aluminum hydroxide (50 μg) + CpG (5 μg) |
72% |
68% |
63% |
|
gE (5 μg) + Aluminum hydroxide (50 μg) + CpG (10 μg) |
67% |
60% |
61% |
Comments 10 : [ Conclusions; 5. Discussion. It is awkward to have the Conclusions before the Discussion. I suggest to switch the order.]
Response 10 : [Agree. We have reversed the order of the conclusions.]
Comments 11 : [ References are still incomplete (missing page numbers), e.g., Refs 3, 14, 15, 19, 22, 24, 32, 34.]
Response 11 : [Agree. We have revised this part of the content, thank you for your correction.]
Thank you for your comments. We will refine the language after the manuscript is initially finalized.
Round 3
Reviewer 2 Report
Comments and Suggestions for Authors
The authors have addressed my previous comments.